# Vanadium(IV) Complexes with Methyl-Substituted 8-Hydroxyquinolines: Catalytic Potential in the Oxidation of Hydrocarbons and Alcohols with Peroxides and Biological Activity

**DOI:** 10.3390/molecules26216364

**Published:** 2021-10-21

**Authors:** Joanna Palion-Gazda, André Luz, Luis R. Raposo, Katarzyna Choroba, Jacek E. Nycz, Alina Bieńko, Agnieszka Lewińska, Karol Erfurt, Pedro V. Baptista, Barbara Machura, Alexandra R. Fernandes, Lidia S. Shul’pina, Nikolay S. Ikonnikov, Georgiy B. Shul’pin

**Affiliations:** 1Institute of Chemistry, University of Silesia, Szkolna 9, 40-006 Katowice, Poland; joanna.palion-gazda@us.edu.pl (J.P.-G.); katarzyna.choroba@us.edu.pl (K.C.); jacek.nycz@us.edu.pl (J.E.N.); 2Associate Laboratory i4HB-Institute for Health and Bioeconomy, NOVA School of Science and Technology, NOVA University Lisbon, 2819-516 Caparica, Portugal; af.luz@campus.fct.unl.pt (A.L.); l.raposo@campus.fct.unl.pt (L.R.R.); pmvb@fct.unl.pt (P.V.B.); 3UCIBIO—Applied Molecular Biosciences Unit, Department of Life Sciences, NOVA School of Science and Technology, NOVA University Lisbon, 2819-516 Caparica, Portugal; 4Faculty of Chemistry, University of Wroclaw, F. Joliot-Curie 14, 50-383 Wroclaw, Poland; alina.bienko@chem.uni.wroc.pl (A.B.); agnieszka.lewinska@chem.uni.wroc.pl (A.L.); 5Department of Chemical Organic Technology and Petrochemistry, Silesian University of Technology, Krzywoustego 4, 44-100 Gliwice, Poland; karol.erfurt@polsl.pl; 6A.N. Nesmeyanov Institute of Organoelement Compounds, Russian Academy of Sciences, Ulitsa Vavilova 28, 119991 Moscow, Russia; shulpina@ineos.ac.ru (L.S.S.); ikonns@ineos.ac.ru (N.S.I.); 7N.N Semenov Federal Research Center for Chemical Physics, Russian Academy of Sciences, Ulitsa Kosygina 4, 119991 Moscow, Russia; 8Chair of Chemistry and Physics, Plekhanov Russian University of Economics, Stremyannyi Pereulok 36, 117997 Moscow, Russia

**Keywords:** vanadium(IV) complexes, biological activity, catalytic properties, 8-hydroxyquinoline, cytotoxicity studies

## Abstract

Methyl-substituted 8-hydroxyquinolines (Hquin) were successfully used to synthetize five-coordinated oxovanadium(IV) complexes: [VO(2,6-(Me)_2_-quin)_2_] (**1**), [VO(2,5-(Me)_2_-quin)_2_] (**2**) and [VO(2-Me-quin)_2_] (**3**). Complexes **1**–**3** demonstrated high catalytic activity in the oxidation of hydrocarbons with H_2_O_2_ in acetonitrile at 50 °C, in the presence of 2-pyrazinecarboxylic acid (PCA) as a cocatalyst. The maximum yield of cyclohexane oxidation products attained was 48%, which is high in the case of the oxidation of saturated hydrocarbons. The reaction leads to the formation of a mixture of cyclohexyl hydroperoxide, cyclohexanol and cyclohexanone. When triphenylphosphine is added, cyclohexyl hydroperoxide is completely converted to cyclohexanol. Consideration of the regio- and bond-selectivity in the oxidation of n-heptane and methylcyclohexane, respectively, indicates that the oxidation proceeds with the participation of free hydroxyl radicals. The complexes show moderate activity in the oxidation of alcohols. Complexes **1** and **2** reduce the viability of colorectal (HCT116) and ovarian (A2780) carcinoma cell lines and of normal dermal fibroblasts without showing a specific selectivity for cancer cell lines. Complex **3** on the other hand, shows a higher cytotoxicity in a colorectal carcinoma cell line (HCT116), a lower cytotoxicity towards normal dermal fibroblasts and no effect in an ovarian carcinoma cell line (order of magnitude HCT116 > fibroblasts > A2780).

## 1. Introduction

In the last three decades, vanadium coordination compounds have received increasing interest due to their structural features [1,2,3,4,5,6,7,8,9,10,11,12,13,14,15,16,17,18,19,20,21,22,23,24,25,26,27,28,29,30,31,32,33,34,35,36,37,38,39,40,41,42,43,44], catalytic applications [21,22,23,24,25,26,27,28,29,30,31,32,33,45,46] and medicinal importance [33,34,35,36,37,38,39,40,41,42,43,44,47,48,49,50,51,52,53,54,55,56,57,58,59,60,61,62,63]. Particular attention has been paid to bis-chelated V^IV^O and V^V^O complexes of the general formula [V^IV^O(N∩O)_2_], [V^IV^O(N∩O)_2_(solvent)] and [V^V^O(OR)(N∩O)_2_]. Among them there is the vanadium maltolate complex [VO(ethylmaltolate)2] which entered phase IIa clinical trials as an antidiabetic agent [56]. For V^IV^O complexes bearing bidentate picolinate ligands, it was found that their pharmacological potential is strongly dependent on the structural modification of organic ligands and ligand arrangement around the metal center [54,57,58,59,60,61]. Exemplarily, the introduction of an electron-withdrawing halogen atom or electron-donating alkyl group at the fifth or third position of the pyridine ring leads to stronger insulin-enhancing activity of [VO(picolinate)_2_] and [VO(picolinate)_2_(solvent)] in comparison with [VO(picolinate)_2_(H_2_O)]. On the contrary, investigations of the substituent effect on the antiproliferative potential of vanadium complexes [V^V^O(OMe)(N∩O)_2_] bearing 8-hydroxyquinoline ligands on HCT116 and A2780 cancer cell lines, showed that the introduction of substituents into the 8-hydroxyquinoline backbone at 5- and 5,7-positions induces a reduction of the antiproliferative effect in relation to [VO(OMe)(quin)_2_], and the complexes [V^V^O(OMe)(N∩O)_2_] with hydroxyquinoline (quin) substituted only in the 5-position were more cytotoxic than those with substituents in the 5,7-positions of the quin backbone. Nevertheless, as all the vanadium(V) complexes [V^V^O(OR)(N∩O)_2_] with 8-hydroxyquinoline derivatives showed a significantly lower IC_50_ towards the A2780 cell line, other than oxovanadium and dioxovanadium complexes previously reported [62], these systems deserve further intensive studies. What is also important, is that monomeric oxidovanadium(V) complexes [VO(OMe)(N∩O)_2_] with nitro- or halogen-substituted quinolin-8-olate ligands were found to be very promising in view of their catalytic properties. These complexes exhibit high catalytic activity toward the oxidation of inert alkanes to alkyl hydroperoxides by H_2_O_2_ in aqueous acetonitrile, with the yield of oxygenate products up to 39% and a TON of 1730 for 1 h [63].

In continuation of our studies on monomeric oxovanadium complexes with bidentate monoanionic ligands [62,63], we present herein the biological and catalytic potential of three oxovanadium(IV) complexes with methyl-substituted 8-hydroxyquinolines, they are [VO(2,6-(Me)_2_-quin)_2_] (**1**), [VO(2,5-(Me)_2_-quin)_2_] (**2**) and [VO(2-Me-quin)_2_] (**3**). Two of them (**1** and **2**) have been obtained for the first time, while compound **3**, the X-ray structure of which has been presented previously [64], was included to obtain more reliable structure–activity relationships for these systems. In contrast to the previously reported [V^V^O(OMe)(N∩O)_2_] with 8-hydroxyquinoline derivatives bearing substituents in the 5- or 5,7-positions of the quin backbone, all the complexes here presented contain a vanadium(IV) ion and they are five-coordinated, which was evidenced by the X-ray diffraction analysis, EPR and UV–Vis spectroscopy. The catalytic potential of complexes **1**–**3** was examined for the oxidation of alkanes with H_2_O_2_ and compared with the activity of the classic system vanadate ion plus pyrazine-carboxylic acid (PCA). To evaluate the antiproliferative effect of complexes **1**–**3**, HCT116 and A2780 cancer cell lines, and normal dermal fibroblasts were used.

## 2. Results and Discussion

### 2.1. Synthesis

To synthetize the oxidovanadium complexes with methyl-substituted 8-hydroxyquinolines, [VO(2,6-(Me)_2_-quin)_2_] (**1**), [VO(2,5-(Me)_2_-quin)_2_] (**2**), the previously reported procedure based on the reaction of bis(acetylacetonato)oxidovanadium(IV) with the corresponding 8-hydroxyquinoline derivative in open air was employed [62,63]. Remarkably, in the reaction with the use of quinH, 5-Cl-quinH, 5-NO_2_-quinH, 5,7-Cl_2_-quinH, 5,7-(Me)_2_-quinH, 5,7-Cl,I-quinH and 5,7-I_2_-quinH, 5,7-(Me)_2_-quinH, the vanadium(IV) of the starting [VO(acac)_2_] undergoes oxidation by molecular oxygen and the acetylacetonato ligands are exchanged by the corresponding quinolin-8-olate ions to give six-coordinated [V^V^O(OMe)(N∩O)_2_]. In contrast, the reactions of [VO(acac)_2_] with 2-Me-quinH, 2,6-(Me)_2_-quinH and 2,5-(Me)_2_-quinH resulted in the formation of five-coordinated [V^IV^O(N∩O)_2_]. It indicates that the methyl-functionalization of quinH strengthen its coordination capacity to the V(IV) ion, and steric hindrance induced by the methyl group at the 2-position in the pyridine ring facilitates the formation of five-coordinated complexes.

### 2.2. Molecular Structure

Perspective views of the molecular structures of **1** and **2** together with the atom numbering are depicted in Figure 1. The atoms V(1) and O(2) in the structure **1** are located on a twofold crystallographic axis, thus the molecule [VO(5,6-(Me)_2_-quin)_2_] has crystallographically-imposed twofold symmetry.

The five-coordinated vanadium(IV) ion adopts a coordination geometry lying between square pyramidal and trigonal bipyramidal. The angular structural index parameter τ [65], expressed as the difference between the two largest angles divided by 60, has a value of 0.57 for **1** and 0.56 for **2**. The atom V(1) is shifted ~0.6 Å towards the oxo ion from the least-squares plane defined by the nitrogen and oxygen atoms of two quinolin-8-olate bidentate ligands. The bite angle of the chelating ligand is ~80° (Appendix A), and the dihedral angle between the least-squares planes formed by the organic ligands is equal to 60.38(4)° for **1** and 50.32(3)° for **2**. This kind of coordination geometry seems to be typical for [VO(N∩O)_2_] bearing two N,O-donor bidentate ligands, contrary to [VO(O∩N∩N∩O)] of tetradentate Schiff base ligands in which five-coordinated vanadium(IV) generally displays a distorted square-pyramidal environment with a basal square plane constituted by the two iminic nitrogen and two phenoxide oxygen atoms [4,6,64,66,67,68,69,70,71,72,73,74,75,76,77,78,79,80,81,82] (Appendix A). The V = O [1.590(2) and 1.595(3) Å], V–O [1.921(1) and 1.924(2) Å] and V–N [2.122(2) and 2.125(2) Å] bond lengths in **1** and **2** (Appendix A) are in good agreement with those reported for five-coordinated vanadyl compounds (Appendix A). More detailed structural parameters of the designed complexes are included in Appendix A.

The phase purity of **1** and **2** was evidenced by comparing the powder X-ray diffraction (PXRD) patterns of the powdery sample with those generated by simulation based on single-crystal structures. As shown in Appendix A, the bulk powder samples give patterns consistent with those obtained theoretically from the single-crystal structure.

### 2.3. EPR Spectroscopy

The oxidation state of vanadium in complexes **1**–**3** was confirmed by the EPR spectra (Figure 2). The X- and Q-band EPR powder spectra of all complexes were recorded at room temperature. The X-band spectrum shows intense central field broad structured bands with no detectable hyperfine structure at g = 1.982 for **1**, 1.986 for **2** and 1.996 for **3**. The Q-band spectrum presents slightly resolved peaks, and the nature of the peaks is the same as in the X-band. The 77 K powder EPR spectra recorded at the X-band show that there is a slight increase in the signal intensity.

EPR frozen solution spectra of compounds **1**–**3** in DMSO (Figure 3) and CH_3_CN (Appendix A) show eight lines of hyperfine splitting of a parallel and perpendicular orientation, proving the interaction of S = 1/2 with the nucleus spin of one vanadium and hence the formation of mononuclear compounds. The spectra of these mononuclear compounds may be simulated using the spin Hamiltonian parameters g_x_ = g_y_ = 1.979, g_z_ = 1.942, A_x_ = A_y_ = 65 G, A_z_ = 180 G for **1**, g_x_ = g_y_ = 1.977, g_z_ = 1.958, A_x_ = A_y_ = 52 G, A_z_ = 169 G for **2**, and g_x_ = g_y_ = 1.994, g_z_ = 1.946, A_x_ = A_y_ = 50 G, A_z_ = 169G for **3**, respectively, which are typical for oxidovanadium(IV) compounds with an analogous N_2_O_2_ donor set of the ligands in the vanadium xy plane [38,83,84], and in agreement with the molecular structure determined by X-ray crystal structure studies of **1**–**3**.

In order to investigate the stability of the studied complexes, EPR spectra were recorded depending on the time (Figure 4). The constant position of bands confirms the stability of the V(IV) complexes in the solution.

### 2.4. Absorption Spectroscopy

The yellowish-brown tetravalent *d^1^* complexes **1** and **3** display three ligand-field transitions *d_xy_* → *d_xz_*, *d_yz_* and *d_xy_* → *d*_x_^2^_−y_^2^ and *d_xy_* → *d*_z_^2^ at 701, 585 and 523 nm for **1** and 744, 572 and 501 nm for **3**. In the case of **2**, only two bands of low intensity in the range 500–850 nm, are observed. An expected a third transition attributed to *d_xy_* → *d*_z_^2^ is most likely masked by the intense ligand-metal charge-transfer (LMCT) transition with a maximum at 398 nm. In the spectra of **1** and **3**, the absorptions assigned transitions from the p_π_ orbital of the phenolate oxygen to d_π_ orbitals of the vanadium center (LMCT) occur at 380 and 383 nm, respectively. The higher energy bands of **1**–**3,** are attributable to the spin-allowed ligand centered transitions (IL) π → π * of quinolin-8-olate (Appendix A).

UV–Vis spectroscopy was also used to study the stability of the five-coordinated oxidovanadium(IV) complexes in solution, and UV–Vis spectra of **1**–**3** in DMSO and CH_3_CN (10^−4^ M) were collected once every two hours over 24 h at room temperature. As observed, the main bands remained constant in the electronic spectra, indicating stability of the V(IV) complexes in solution (Figure 5 and Appendix A).

### 2.5. Catalytic Oxidations with Hydrogen Peroxide

One of us discovered in 1993, a system for the efficient oxidation of various organic compounds with hydrogen peroxide, based on a simple vanadate ion compound. The obligatory component of this system was PCA (pyrazinecarboxylic acid). This reaction takes place at low temperatures in a solution of acetonitrile [85]. Further, this system was studied in detail, including the oxidation of alkanes, olefins, arenes, and alcohols with hydrogen peroxide and other oxidizing agents [86,87,88]. In the absence of a catalyst, the reaction proceeds extremely slowly, in five hours the yield of products is no more than 3–5% [89,90,91,92].

In this work, we present a study of the catalytic activity of **1**–**3** and the effect of 2-pyrazinecarboxylic acid on the activities of these complexes. It turned out that all complexes exhibit high catalytic activity in the oxidation of alkanes, but only in the presence of PCA. The curves for the accumulation of oxidation products are shown in Figure 6, Figure 7, Figure 8 and Figure 9.

In the case of using complexes **1**–**3**, the yields of alkane oxidation products are noticeably higher than in reactions catalyzed by the vanadate anion-PCA system (Appendix A). Compared to other vanadium complexes, that we and other researchers have used in the oxidative catalysis of alkanes and alcohols, we can say that the complexes described in this article exhibit a high activity. We have also studied the oxidation of cyclohexane in the presence of nitric acid. In this case, for complex **1**, the yield of oxidation products is much lower than in the reaction using PCA (see Figure 7). And in the case of complexes **2** and **3**, the oxidation reactions of cyclohexane do not lead to any noticeable yields of oxidation products.

Consideration of the regio- and bond-selectivity in the oxidation of n-heptane and methylcyclohexane indicates that the oxidation proceeds with the participation of free hydroxyl radicals. The regio-selectivity parameters for the oxidation of n-heptane were obtained for complex **2**: C (1):C (2):C (3):C (4) = 1.0:5.0:5.0:4.4. The bond-selectivity parameters for the oxidation of methylcyclohexane were also obtained: for complex **1:** 1°:2°:3° = 1.0:6.3:16.7; for complex **2**: 1°:2°:3° = 1.0:5.5:15.0; for complex **3**: 1°:2°:3° = 1.0:5.7:16.0, respectively. These values are close to the parameters obtained for oxidation reactions with the participation of hydroxyl radicals, although they are somewhat higher, apparently due to steric hindrances created by chelating ligands around the catalytic center in the molecules of our complexes [93].

The complexes show moderate activity in the oxidation of alcohols. The yields for the oxidation of phenylethanol to acetophenone with tert-butyl hydroperoxide under catalysis with complexes **1**–**3,** were 46%, 23% and 32%, respectively, at a temperature of 50 °C, in acetonitrile for four hours. Hydrogen peroxide is much less productive in these reactions. In analogous oxidation reactions of cyclohexanol to cyclohexanone, corresponding yields were 10%, 5.5% and 5.5% after 5 h. In the oxidation of cyclohexene, yields of cyclohexene-1-ol were 10%, 6.5% and 5.5% correspondingly for complexes **1**–**3**. Epoxide and other products of oxidation were formed in insignificant quantities.

### 2.6. Viability Studies

HCT116 and A2780 cancer cell lines and normal dermal fibroblasts were used to evaluate the antiproliferative effect of complexes **1**–**3** (Figure 10 and Appendix A and Table 1).

Our results show that complex **1** displayed similar cytotoxicity in both cancer cell lines and in normal dermal fibroblasts (Figure 10 and Appendix A and Table 1). Complex **2** displayed a higher cytotoxicity in normal dermal fibroblasts compared to HCT116 and A2780 tumor cell lines (Figure 10 and Appendix A and Table 1). On the other hand, complex **3** showed a higher cytotoxicity in the HCT116 cell line (5.9 µM) compared to normal dermal fibroblasts and the A2780 tumor cell line (Figure 11 and Appendix A and Table 1). Moreover, HCT116 was the cancer cell model more sensitive to the complexes (Figure 10 and Appendix A and Table 1). Interestingly, all complexes show a higher cytotoxicity (lower IC_50_) compared to cisplatin (IC_50_ of 15 µM; see Appendix A).

The complexes herein described exhibit an IC_50_ in the range of 5.9 to 50.9 µM in HCT116 and A2780 cancer cell lines (Figure 11 and Appendix A and Table 1). Previously published complexes bearing [VO(OMe)(quin)_2_] backbones, possess an IC_50_ between 0.96 and 10.09 µM in HCT116 and A2780 cancer cell lines and from 11.24 to over 100 µM in normal dermal fibroblasts [62]. This may be an indication that complexes bearing VO(2-Me-Quin) backbones are not as promising for therapeutic use as previously described complexes bearing [VO(OMe)(quin)_2_] backbones [62]. Published results of the IC_50_ values of complexes bearing hydroxyquinoline derived ligands in cancer cell lines other than HCT116 and A2780 range from 0.9 to 219 µM, and oxovanadium and dioxidovanadium complexes present IC_50_ values between 0.96 to 224.5 µM, which indicates that the complexes here described have an IC_50_ in the lower end of the range (Table 1) [62,94,95].

To further understand the mechanisms of cytotoxicity associated with exposure to complexes **1**–**3,** additional biological studies in the HCT116 cell line were carried out.

### 2.7. Complex Internalization

Internalization of the complexes was evaluated by exposing HCT116 cancer cells to 20 × IC_50_ concentrations of the complexes **1**–**3** for 3 and 6 h at 4 °C and 37 °C (Figure 11).

Our results show that temperature does induce statistically significant alterations in the amount of complex internalized, which suggests that there is no active transport of complexes into cells after 3 or 6 h (Figure 11), which has also been described in the literature with oxovanadium complexes [96]. After 6 h incubation, approximately 94% and 91% of complexes **2** and **3**, respectively, and 60% of complex **1** were found in the HCT116 cellular fraction (Figure 11). These results suggest that the cytotoxicity might be directly correlated with the amount of internalized complex over time (Figure 10 and Figure 11) and complexes **2** and **3** appear to be retained within HCT116 cells (Figure 11).

### 2.8. Induction of Apoptosis in the HCT116 Cell Line Exposed to Complexes ***1***–***3***

To understand if the loss of cellular viability induced by exposure to complexes **1**–**3** is associated with the triggering of programed cell death, HCT116 cells were exposed to the respective IC_50_ concentrations for 48 h, and the Annexin V-Alexa fluor 488/PI dead cell apoptosis assay (ThermoFisher, Waltham, MA, USA) was used to determine the percentage of apoptotic cells (Figure 12).

Our results show a statistically significant increase in the percentage of apoptotic cells due to exposure to complexes **1**–**3** (Figure 12). HCT116 cells displayed 14.5%, 12.6% and 10.4% of apoptotic cells after 48 h exposure to complexes **1**–**3**, respectively, while the DMSO condition presented only 7.3% of apoptotic cells (Figure 12). There is also an increase in necrotic cells when compared with the DMSO control, which has 2% of necrotic cells, in cells exposed to complexes **1**–**3**, of 7.0%, 6.2% and 5.0%, respectively. All complexes induced lower apoptotic cell death when compared with cisplatin (Figure 12). The induction of apoptosis has been reported for other vanadium complexes bearing 8-hydroxyquinoline-based ligands and dioxidovanadium(V) complexes [62,95]. Based on the viability data (Figure 10 and Appendix A and Table 1) and internalization data (Figure 11) we were expecting a higher cell death for complex **2** and particularly for complex **3**. Since we did not observe this trend, additional cell death mechanisms might be involved in the loss of cell viability. To further explain this issue, we assessed autophagic HCT116 cell death after exposure to complexes **1**–**3**.

### 2.9. Induction of Autophagy in the HCT116 Cell Line Exposed to Complexes ***1***–***3***

Another mechanism of cell death triggered by exposure to metallic complexes is autophagy. The Autophagy Assay kit (Abcam) was used to determine the percentage of autophagic cells present after 48 h exposure to IC_50_ concentrations of complexes **1**–**3** (Figure 13).

Exposure to complexes **1**–**3** led to a 7×, 4× and 10× increase of autophagic cells, respectively when compared to the DMSO control (Figure 13). Interestingly, complex **3** was able to induce more autophagic cell death compared to cisplatin (but to a lesser extent compared to rapamycin) (Figure 13). Therefore, induction of apoptosis and autophagy are responsible for the total loss of cell viability induced by the complexes in HCT116 cells (Figure 10, Figure 11 and Figure 13 and Table 1).

### 2.10. Intracellular Reactive Oxygen Species (ROS) Production in the HCT116 Cell Line Exposed to Complexes ***1***–***3***

ROS are known triggers of apoptosis and autophagy in cells exposed to metallic complexes. ROS production was evaluated in HCT116 cells exposed to IC_50_ concentrations of complexes **1**–**3** for 48 h (Figure 14).

Our results show a statistically significant increase of 3.3× and 2.3× of intracellular ROS in HCT116 cells exposed to complexes **2** and **3**, respectively when compared to the DMSO control, although lower than cisplatin (Figure 14). Although not statistically significant, complex **1** also induces a 1.15× increase in the amount of intracellular ROS in HCT116 cells (Figure 14).

### 2.11. Evaluation of Alterations in the Mitochondrial Membrane Potential of HCT116 Cells Exposed to Complexes ***1***–***3***

To evaluate if the observed apoptosis induction in the HCT116 cell line is triggered by the intrinsic pathway, due to destabilization of mitochondria, changes in the mitochondrial membrane potential were evaluated using JC-1 dye (Abnova). This is a green fluorescent dye as a monomer; however, it will aggregate in the presence of a normal mitochondrial potential, which will cause a red-shift in emission spectra [97]. Measuring the green and red fluorescence by flow cytometry it is possible to calculate normalized ratios that can show us if there is an increase or decrease in the mitochondrial membrane potential (Figure 15).

Our results shown that there is no significant change of the mitochondrial membrane potential in HCT116 cells exposed to complexes **1**–**3** when compared with the DMSO control (Figure 15). The cisplatin control displays a small increase in the JC-1 (monomer/aggregate) ratio, which is indicative of mitochondrial membrane destabilization, although not statistically significant (Figure 15).

## 3. Materials and Methods

### 3.1. Materials

2-methyl-8-hydroxyquinoline was commercially available (Sigma Aldrich, Darmstadt, Germany), while 2,5-dimethyl-8-hydroxyquinoline and 2,6-dimethyl-8-hydroxyquinoline were synthesized according to procedures described previously [98]. The analytical data (FT-IR and HR-MS spectra, X-ray analysis and PXRD spectrum) for [VO(2-Me-quin)_2_] (**3**) provided a good agreement with those reported in reference [64].

### 3.2. Synthesis of [VO(2,6-(Me)_2_-quin)_2_] (***1***) and [VO(2,5-(Me)_2_-quin)_2_] (***2***)

A mixture of VO(acac)_2_ (1 mmol, 0.26 g) and the appropriate 2,5-dimethyl-8-hydroxyquinoline or 2,6-dimethyl-8-hydroxyquinoline (2 mmol; 0.35 g) was suspended in toluene (30 mL), and the resulting solution was refluxed for 2 h in open air. After several days the crystalline solid of the vanadium(IV) complex was collected. The crystals suitable for X-ray analysis were obtained by recrystallization from an acetonitrile/chloroform mixture (1/1 *v*/*v*).

[VO(2,6-(Me)_2_-quin)_2_] (**1**): Yield 55%. HR-MS (ESI): calcd for C_22_H_20_N_2_O_3_NaV^+^ [M + Na]^+^ 434.0811, found 434.0818. Anal. Calc. for C_22_H_20_N_2_O_3_V (411.34 g/mol): C 64.25, H 4.90, N 6.81%. Found: C 64.55, H 4.73, N 6.78%. IR (KBr, cm^−1^): 3436(s), 3028(w), 2920(m), 1613(w), 1575(s), 1496(s), 1469(m), 1439(w), 1401(s), 1377(w), 1346(m), 1266(s), 1208(w), 1041(s), 1132(s), 1033(w), 987(m), 976 (m), 946(s), 847(m), 789(w), 782(w), 725(m), 686(m), 653(m), 634(m), 618(w), 600(w), 533(m), 432(w). UV–Vis (DMSO, λ_max_, nm (ε, dm^3^·mol^−1^·cm^−1^)): 268 (25170), 312 (1930), 380 (2310), 523 (530), 585 (280), 701 (110).

[VO(2,5-(Me)_2_-quin)_2_] (**2**): Yield 75%. HRMS (ESI): calcd for C_22_H_20_N_2_O_3_NaV^+^ [M + Na]^+^ 434.0811, found 434.0813. Anal. Calc. for C_22_H_20_N_2_O_3_V (411.34 g/mol): C 64.25, H 4.90, N 6.81%. Found: C 63.94, H 4.69, N 6.52%. IR (KBr, cm^−1^): 3438(m), 3028(m), 2892(w), 2861(w), 1597(m), 1571(s), 1515(s), 1462(s), 1434(s), 1409(m), 1374(m), 1314(s), 1291(m), 1263(s), 1245(m), 1223(w), 1157(m), 1093(s), 983(s), 947(w), 836(s), 819(w), 787(m), 767(s), 747(m), 644(m), 632(m), 523(m), 501(w), 452(w). UV–Vis (DMSO, λ_max_, nm (ε, dm^3^·mol^−1^·cm^−1^)): 320 (3960), 398 (6600), 582 (100), 753 (80).

### 3.3. X-ray Crystal Structure Determination

The X-ray diffraction data for complexes **1** and **2** were collected using an Oxford Diffraction four-circle diffractometer Gemini A Ultra with an Atlas CCD detector using graphite monochromated MoKα radiation (λ = 0.71073 Å) at room temperature. Diffraction data collection, cell refinement and data reduction were performed using the CrysAlis^Pro^ software [99]. The structures were solved by direct methods using SHELXS and refined by full-matrix least-squares on *F*^2^ using SHELXL-2014 [100]. All the non-hydrogen atoms were refined anisotropically, and hydrogen atoms were placed in calculated positions and refined with riding constraints: *d*(C–H) = 0.93 Å, *U*_iso_(H) = 1.2 *U*_eq_(C) (for aromatic) and *d*(C–H) = 0.96 Å, *U*_iso_(H) = 1.5 *U*_eq_(C) (for methyl). The methyl groups were allowed to rotate about their local threefold axis. Details of the crystallographic data collection, structural determination, and refinement for **1** and **2** are given in Table 2, whereas selected bond lengths and angles for them are listed in Appendix A.

Powder X-ray diffraction (XRPD) measurements on **1** and **2** were performed on a PANalytical Empyrean X-ray diffractometer by using Cu−K_α_ radiation (λ = 1.5418 Å), in which the X-ray tube was operated at 40 kV and 30 mA ranging from 5 to 50° (Figure 2 and Appendix A).

### 3.4. Physical Measurements

IR spectra were recorded on a Nicolet iS5 FT–IR spectrophotometer in the spectral range 4000–400 cm^−1^ with the samples in the form of KBr pellets (Appendix A). HRMS analysis (Appendix A) was performed on a Waters Xevo G2 Q-TOF mass spectrometer (Waters Corporation, Milford, MA, USA) with an ESI ion source. Full-scan MS data were collected from 100 to 1000 Da in positive ion mode with a scan time of 0.5 s. To ensure accurate mass measurements, data were collected in centroid mode and mass was corrected during acquisition using a leucine enkephalin solution as an external reference (Lock-Spray^TM^), with the reference ion at *m*/*z* 556.2771 Da ([M + H]^+^). Elemental analyses (C, H, N) were performed on a Perkin–Elemer CHN–2400 analyzer. The electronic spectra were obtained using Nicolet Evolution 220 in the range 240–1000 nm in DMSO (Appendix A).

### 3.5. EPR Spectroscopy

Electron paramagnetic resonance (EPR) spectra of the oxidovanadium(IV) complexes were measured using a Bruker ELEXYS E 500 operating at the X-band frequency (9.7 GHz). The solid compounds **1** and **2** dissolved in water and a few drops of DMSO, were added to the samples to ensure good glass formation at liquid nitrogen temperature [101]. A microwave frequency of 6.231 GHz, power of 10 mW and modulation amplitude of 8 G was used. Anisotropic spectra were recorded on frozen solutions at 77 K using quartz Dewar and glass capillary tubes at room temperature. An analysis of the EPR spectra was carried out using the WinEPR SimFonia software package, version 1.26b [102].

### 3.6. Biological Assays

#### 3.6.1. Cell Culture

The human colorectal carcinoma derived cancer cell line (HCT116) and human normal dermal fibroblasts (Ref. PCS-201-010) were purchased from American Type Culture Collection (ATCC, Manassas, VA, USA) and grown in Dulbecco’s modified Eagle medium (DMEM). The human ovarian carcinoma derived cancer cell line (A2780) was purchased from Merck (Darmstadt, Germany) and cultivated in Roswell Park Memorial Institute (RPMI) 1640 culture medium. All media were supplemented with 10% fetal bovine serum and a 1% Pen/Strep solution (all media and supplements were from Thermo Fischer Scientific, Waltham, MA, USA). Cell cultures were maintained at 37 °C, in a humified atmosphere of 5% (*v*/*v*) CO_2_ [103,104].

#### 3.6.2. Viability Assays

Normal dermal fibroblasts and cancer cell lines HCT116 and A2780 were seeded in 96-well plates with a density of 7500 cells per well. After 24 h, culture media was replaced, and cells were exposed to different concentrations of complexes **1**–**3** or DMSO 0.1% (*v*/*v*) (vehicle control) or cisplatin (positive control) for 48 h (Appendix A). Cells exposed to cisplatin were used as a positive control [103,104]. After 48 h of incubation, the CellTiter 96^®^ Aqueous Non-Radioactive Proliferation assay (Promega, Madison, WI, USA) was used to determine cellular viability through the production of formazan through the reduction of 3-(4,5-dimethylthiazol-2-yl)-5-(3-carboxymethoxyphenyl)-2-(4-sulfophenyl)-2H-tetrazolium, inner salt (MTS) by dehydrogenases present in metabolically active cells [103,104]. The amount of formazan can be determined by its absorbance at 492 nm in an Infinite M200 microplate reader (Tecan, Mannedorf, Switzerland) [103,104]. The biological activity of the complexes was compared using the half maximal inhibitory concentration of cellular proliferation (IC_50_) determined with Prism 8.2.1 software for windows (GraphPad Software, La Jolla, CA, USA).

#### 3.6.3. Vanadium Detection in the HCT116 Cell Line by ICP-AES

The internalization of the vanadium complexes in HCT116 cells was evaluated by an inductive plasma atomic emission spectrometry technique (ICP-AES). HCT116 cells were seeded in 25 cm^2^ culture flasks with a cellular density of 1 × 10^5^ cells per mL. After 24 h of incubation at 37 °C in a humified atmosphere of 5% (*v*/*v*) CO_2_, the culture medium was replaced with fresh culture medium containing a 20 × IC_50_ at 48 h concentration of the complexes **1**–**3** or 0.1% (*v*/*v*) of DMSO (vehicle control) followed by incubations of 3 h or 6 h at 37 or 4 °C. We used this concentration to ensure that we were above the detection limit of the technique. This concentration (20 × IC_50_) does not induce a loss of cell viability for the selected time points. The culture medium was then collected, and cells detached with trypsin and centrifuged at 700× *g* for 5 min at 15 °C. The supernatant (culture medium plus trypsin) and the pellet (cells) were stored separately at −20 °C until freshly made *aqua regia* was added to all the samples as previously described [104]. The vanadium quantification in each fraction was performed by ICP-AES, as a paid service.

#### 3.6.4. Evaluation of Apoptosis Induction in the HCT116 Cell Line by Flow Cytometry

Apoptosis induction in HCT116 cells was evaluated using the Annexin V-Alexa fluor 488/PI dead cell apoptosis assay (Thermo Fischer Scientific). Briefly, HCT116 cells were seeded in 6-well plates (2 × 10^5^ cells per well) and then incubated 48 h with the IC_50_ concentrations of complexes **1**–**3** at 37 °C in a humified atmosphere of 5% (*v*/*v*) CO_2_. Cells were also incubated with DMSO 0.1% (vehicle control) and cisplatin 15 µM (positive control). Following the manufacturer’s instructions, after this incubation period, cells were detached with trypsin, washed with PBS 1× and incubated 15 min at room temperature with Annexin V-Alexa fluor 488 assay solution and 10 μg mL^−1^ propidium iodide [97,104]. An Attune acoustic focusing cytometer (ThermoFisher Scientific, Waltham, MA, USA) was used to analyze cells and the resulting information was processed with the respective Attune Cytometric Software 2.1 (ThermoFisher Scientific, Waltham, MA, USA).

#### 3.6.5. Autophagy Induction Evaluation in the HCT116 Cell Line by Flow Cytometry

Autophagy induction in HCT116 cells was evaluated using the Autophagy Assay kit (Abcam), according to the manufacturer’s instructions. HCT116 cells were seeded in 6-well plates at a cellular density of 2 × 10^5^ cells per well. After 24 h, cells were incubated with the IC_50_ concentrations of complexes **1**–**3** for 48 h at 37 °C in a humified atmosphere of 5% (*v*/*v*) CO_2_. In addition to the DMSO 0.1% *v/v* vehicle control, cisplatin 15 µM and rapamycin 0.5 µM were performed as positive controls. After 48 h of incubation, cells were detached from the wells with trypsin and washed with Assay Buffer 1× before being incubated 30 min at 37 °C in DMEM medium with the Green Stain solution. Cells were then washed and resuspended in Assay Buffer 1× and this was followed by analysis in an Attune acoustic focusing cytometer (ThermoFisher Scientific), and the results were analyzed with the respective instrument software (Attune Cytometric Software, version 2.1).

#### 3.6.6. Intracellular Reactive Oxygen Species (ROS) Production Evaluation in the HCT116 Cell Line by Flow Cytometry

The induction of ROS in HCT116 cells was evaluated indirectly by flow cytometry using a specific dye, 2′,7′-dichlorodihydrofluorescein diacetate (H_2_DCF-DA) (Thermo Fischer Scientific). In the presence of ROS, intracellular esterases remove the acetate groups of the dye which leads to an increased fluorescence [97,104]. HCT116 cells were seeded in 6-well plates (4 × 10^5^ cells per well) for the initial 24 h of incubation. Cells were incubated with DMSO 0.1% (*v*/*v*) (vehicle control), 50 µM H_2_O_2_ and 15 µM cisplatin (positives controls) and the complexes **1**–**3** at their IC_50_ concentrations for 48 h at 37 °C in a humified atmosphere of 5% (*v*/*v*) CO_2_. Cells were then detached from the wells with trypsin and washed with PBS 1× before incubation with 100 μM of H_2_DCF-DA for 20 min at 37 °C, then processed in an Attune acoustic focusing cytometer (ThermoFisher Scientific), with the resulting data analyzed using the respective software (Attune Cytometric Software, version 2.1).

#### 3.6.7. Mitochondrial Membrane Potential Evaluation in the HCT116 Cell Line by Flow Cytometry

The mitochondrial membrane potential in HCT116 cells was evaluated using 5,5′,6,6′-Tetrachloro-1,1′,3,3′-tetraethylbenzimidazolocarbocyanine iodide, JC-1 (Abnova Corporation, Walnut, CA, USA). The mitochondrial potential is an important parameter of mitochondrial function, and is thus used as an indicator of cell health. In healthy cells with high potential, JC-1 shows aggregates with intense red fluorescence. On the other hand, in cells with low mitochondrial membrane potential (for example cells in apoptosis), JC-1 remains in the monomeric form, which exhibits green fluorescence^104^ [105]. Briefly, HCT116 cells were seeded in 6-well plates at a cellular density of 2 ×10^5^ cells per well and then incubated 48 h with the IC_50_ concentrations of vanadium complexes **1**–**3** at 37 °C in a humified atmosphere of 5% (*v*/*v*) CO_2_, having as controls DMSO 0.1% (vehicle control), cisplatin 15 µM and doxorubicin 0.4 µM (positive controls). Afterwards cells were detached with trypsin, washed with PBS 1× and incubated 20 min at 37 °C in DMEM medium with the JC-1 solution. Cells were again washed and resuspend in DMEM medium without phenol red and analyzed in an Attune acoustic focusing cytometer (ThermoFisher Scientific).

### 3.7. Catalytic Studies

The total volume of the reaction solution was 5 mL (Caution**:** the combination of air or molecular oxygen and H_2_O_2_ with organic compounds at elevated temperatures may be explosive!). Cylindrical glass vessels with vigorous stirring of the reaction mixture were used for the oxidation of alkanes with hydrogen peroxide which were typically carried out in air in a thermostated solution. Initially, a portion of 50% aqueous solution of hydrogen peroxide was added to the solution of the catalyst and substrate in acetonitrile. The aliquots of the reaction solution were analyzed by GC (a 3700 instrument, fused silica capillary column FFAP/OV-101 20/80 *w*/*w*, 30 m × 0.2 mm × 0.3 µm; argon as a carrier gas. Attribution of peaks was made by comparison with chromatograms of authentic samples). Usually samples were analyzed twice, i.e., before and after the addition portion by portion of the excess of solid PPh_3_. This method was proposed and used by one of us previously [106,107].

Alkyl hydroperoxides are transformed in the GC injector into a mixture of the corresponding ketone and alcohol. Due to this, we quantitatively reduced the reaction samples with PPh_3_ to obtain the corresponding alcohol. This method allows us to calculate the real concentrations not only of the hydroperoxide, but of the alcohols and ketones present in the solution at a given moment. An example is shown in Figure 16.

## 4. Conclusions

The studies revealed that methyl-functionalization of 8-hydroxyquinoline facilitates the formation of five-coordinated oxidovanadium(IV) complexes in the reaction of bis(acetylacetonato)oxidovanadium(IV) 2,6-(Me)_2_-quin, 2,5-(Me)_2_-quin and 2-Me-quin.

Complexes **1**–**3** catalyze very efficient transformation of saturated hydrocarbons into alkyl hydroperoxides, alcohols and ketones. The reaction requires addition of PCA and occurs with the participation of hydroxyl radicals.

Here we show that complex **3** effectively reduced the viability of a HCT116 colorectal cancer cell line with low or no cytotoxicity in normal dermal fibroblasts or in an ovarian carcinoma cell line, respectively. On the other hand, complexes **1** and **2** bearing additional methyl groups show lower antiproliferative activities in HCT116 cells, and complex **2** shows some degree of cytotoxicity towards primary fibroblasts. Complexes internationalization is probably associated with passive transport, with more than 90% of complexes **2** and **3** being in the cellular fraction after a 6 h incubation. All complexes can increase in the production of intracellular ROS which can trigger apoptosis and autophagy in the HCT116 cell line. Mitochondrial membrane potential was not significantly altered by the three complexes in HCT116 cells which may indicate that the intrinsic pathway is not activated.

## Figures and Tables

**Figure 1 molecules-26-06364-f001:**
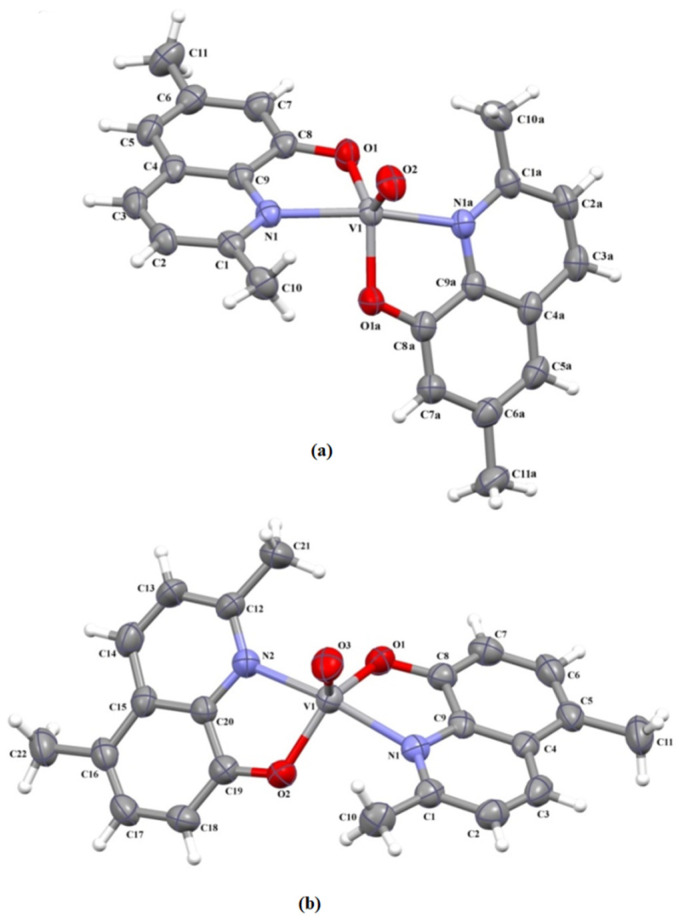
The molecular structures of **1** [symmetry code: (**a**) = −*x*, y, 1/2 − *z*] (**a**) and **2** (**b**). Displacement ellipsoids are drawn at the 50% probability level.

**Figure 2 molecules-26-06364-f002:**
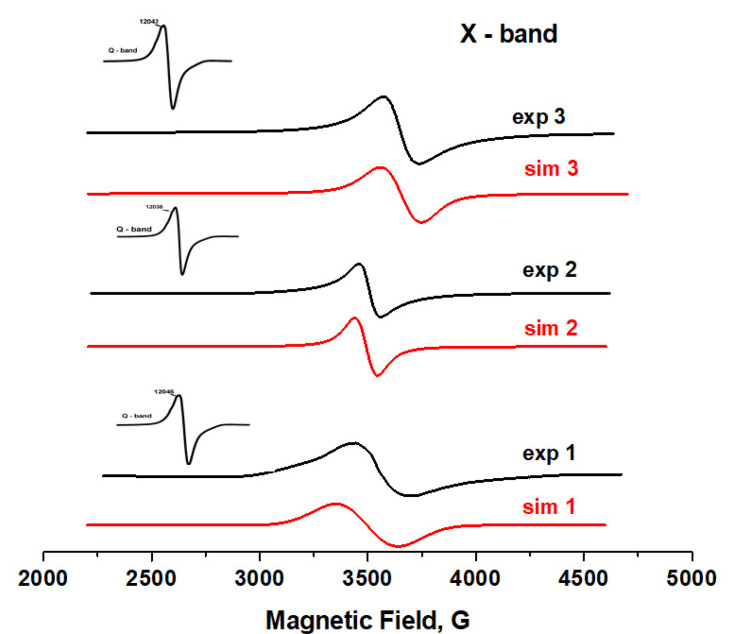
The X-band EPR spectra of **1**–**3** at 77 K together with the spectrum calculated by computer simulation of the experimental spectra with spin Hamiltonian parameters given in the text.

**Figure 3 molecules-26-06364-f003:**
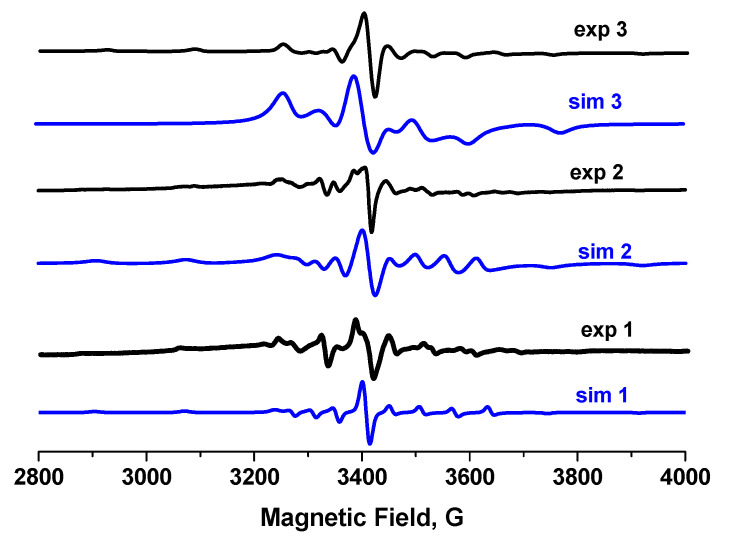
EPR frozen solution spectra (at 77 K) of compounds **1**–**3**; in aqueous 2% DMSO.

**Figure 4 molecules-26-06364-f004:**
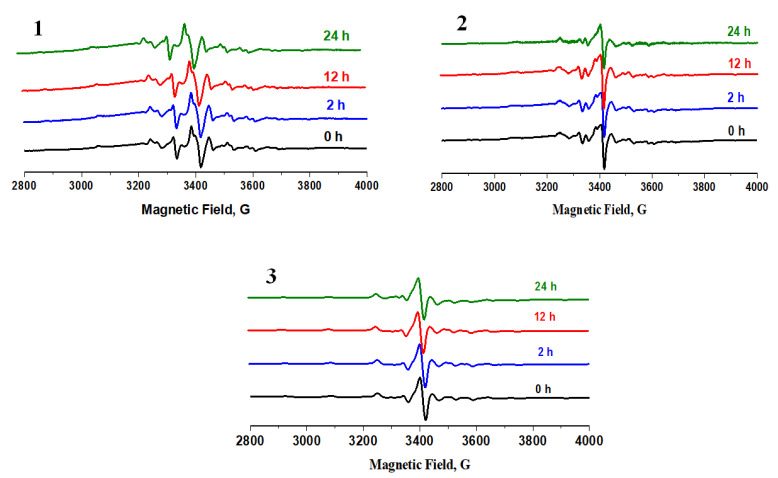
EPR stability spectra (frozen solution at 77 K in aqueous 2% DMSO) of compounds **1**–**3**. Spectra were recorded every for 24 h.

**Figure 5 molecules-26-06364-f005:**
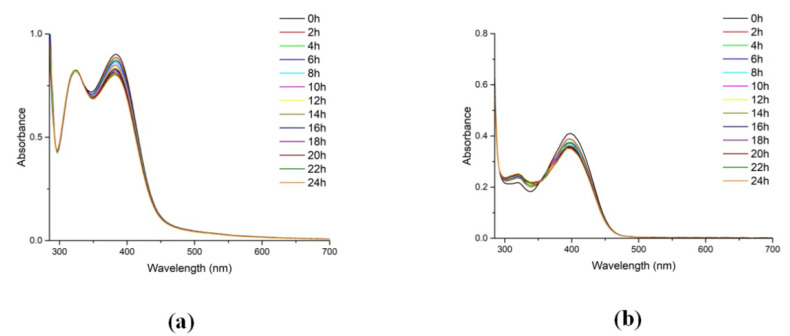
UV–Vis stability spectra in DMSO (10^−4^ M) for compounds **1** (**a**) and **2** (**b**). Spectra were recorded every 2 h for 24 h.

**Figure 6 molecules-26-06364-f006:**
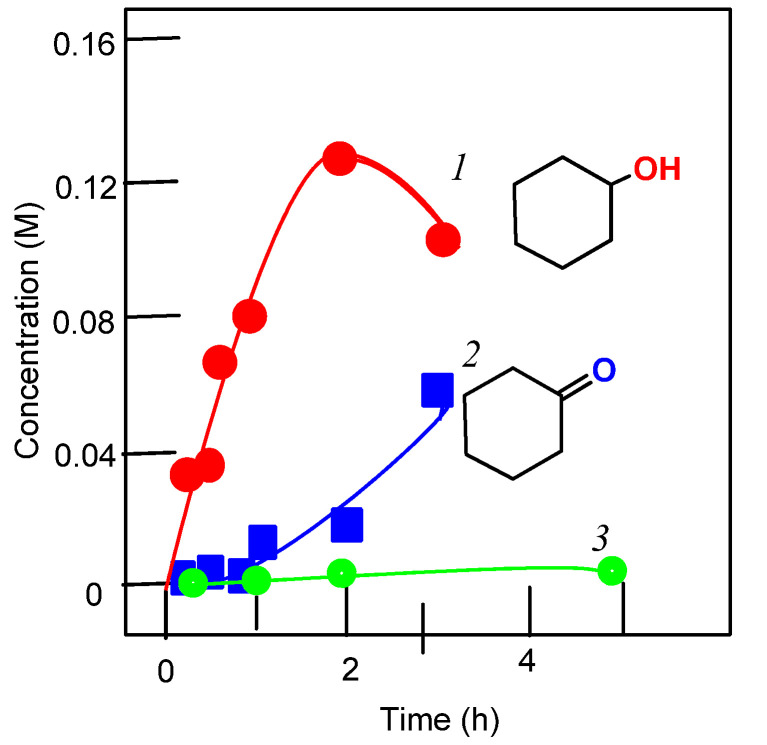
Oxidation of cyclohexane to cyclohexanol (curves 1 and 3) and cyclohexanone (curve 2) with hydrogen peroxide catalyzed by compound **1** in the presence of PCA (curves 1 and 2) and in the absence of PCA (curve 3). Conditions: cyclohexane (0.46 M); H_2_O_2_ (2.0 M, 50% aqueous); complex **1** (5 × 10^−4^ M); PCA (2 × 10^−3^ M) in MeCN at 50 °C. Concentrations of cyclohexanone and cyclohexanol were measured after reduction of the aliquots with solid PPh_3_.

**Figure 7 molecules-26-06364-f007:**
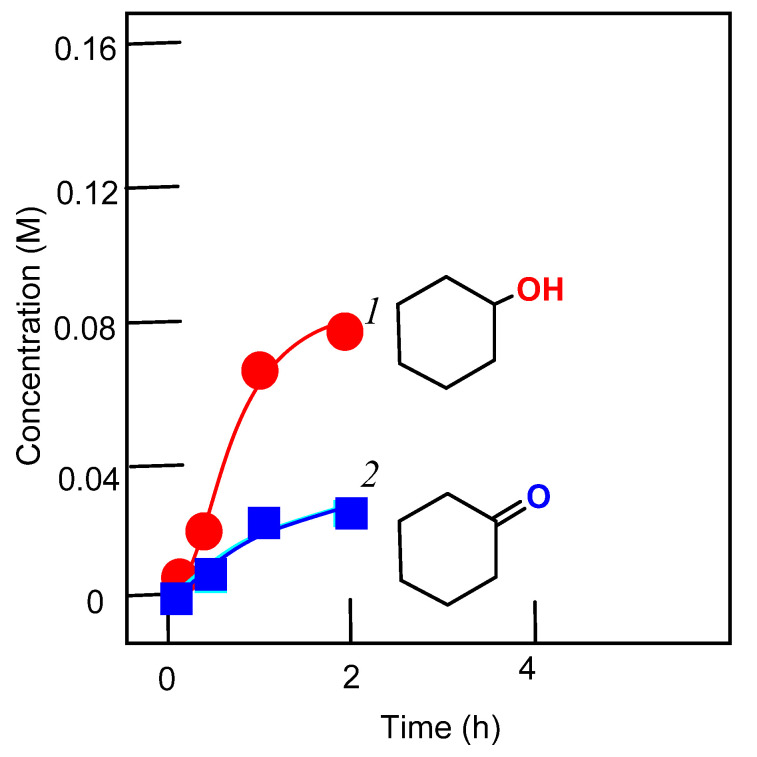
Oxidation of cyclohexane to cyclohexanol (curves 1) and cyclohexanone (curve 2) with hydrogen peroxide catalyzed by compound **1** in the presence of HNO_3_. Conditions: cyclohexane (0.46 M); H_2_O_2_ (2.0 M, 50% aqueous); complex **1** (5 × 10^−4^ M); HNO_3_ (0.05 M) in MeCN at 50 °C. Concentrations of cyclohexanone and cyclohexanol were measured after reduction of the aliquots with solid PPh_3_.

**Figure 8 molecules-26-06364-f008:**
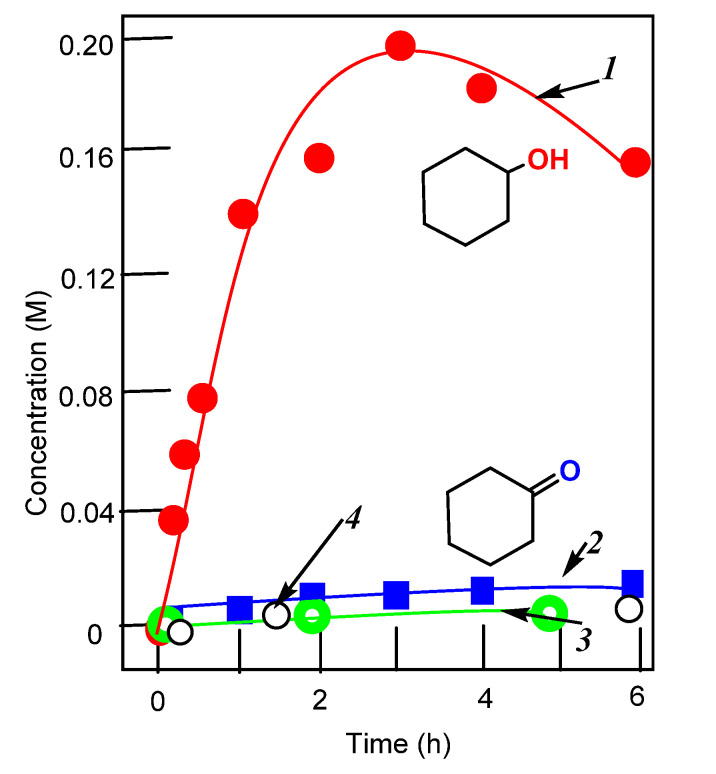
Oxidation of cyclohexane to cyclohexanol (curves 1 and 3) and cyclohexanone (curve 2) with hydrogen peroxide catalyzed by compound **2** in the presence of PCA (curves 1 and 2) and in the absence of PCA (curve 3); symbols marked by the number 4 show the reaction carried out in the absence of PCA and HNO_3_. Conditions: cyclohexane (0.46 M); H_2_O_2_ (2.0 M, 50% aqueous); complex **2** (5 × 10^−4^ M); PCA (2 × 10^−3^ M), HNO_3_ (0.05 M) in MeCN at 50 °C. Concentrations of cyclohexanone and cyclohexanol were measured after reduction of the aliquots with solid PPh_3_.

**Figure 9 molecules-26-06364-f009:**
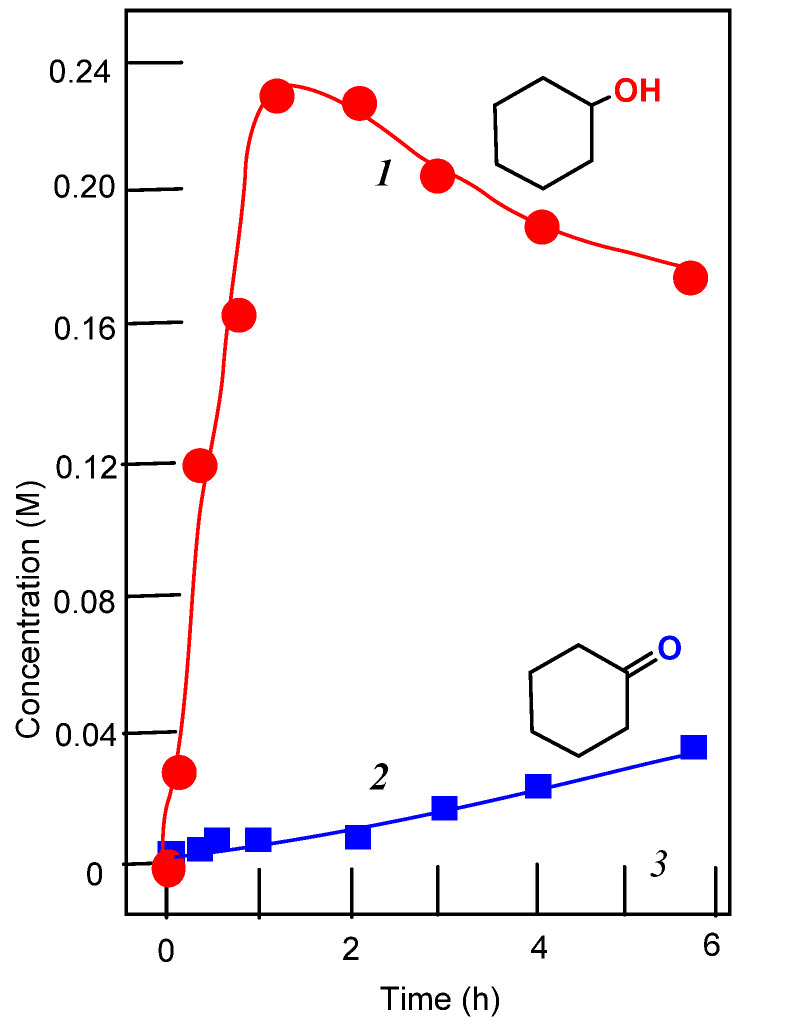
Accumulation of cyclohexanol (curve 1) and cyclohexanone (curve 2) in the oxidation of cyclohexane (0.46 M) with hydrogen peroxide (2.0 M, 50% aqueous) catalyzed by compound **3** (5 × 10^−4^ M); PCA (2 × 10^−3^ M) in MeCN at 50 °C. Concentrations of cyclohexanone and cyclohexanol were measured after reduction of the aliquots with solid PPh_3_. The yield of cyclohexane oxidation products was 48%.

**Figure 10 molecules-26-06364-f010:**
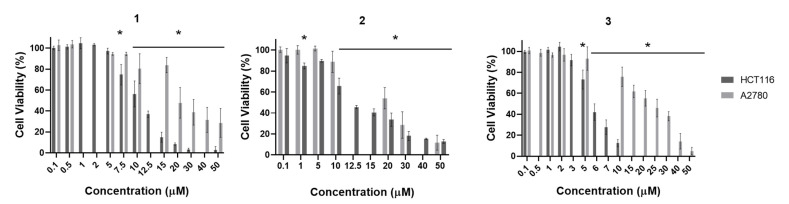
Antiproliferative effect of complexes **1**–**3** in the HCT116 and A2780 cancer cell lines, after 48 h, evaluated by the MTS method. Cell viabilities were normalized to DMSO 0.1% (*v*/*v*) (vehicle control). The results presented are mean ± standard deviation of three independent assays. An asterisk indicates a *p*-value inferior to 0.05.

**Figure 11 molecules-26-06364-f011:**
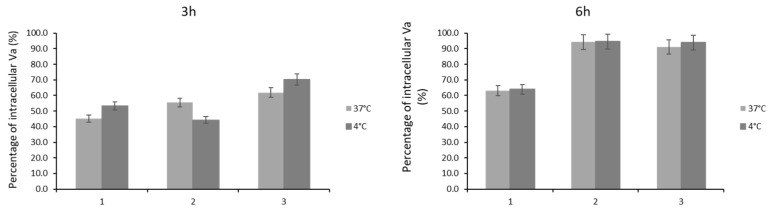
Internalization of complexes evaluated by the determination of the amount of vanadium (determined by ICP-AES) present in the cellular fraction of HCT116 after exposure of HCT116 cells to 20 × IC_50_ concentrations of complexes **1**–**3** for 3 and 6 h at 4 °C and 37 °C. The results presented are mean ± standard deviation of three independent assays.

**Figure 12 molecules-26-06364-f012:**
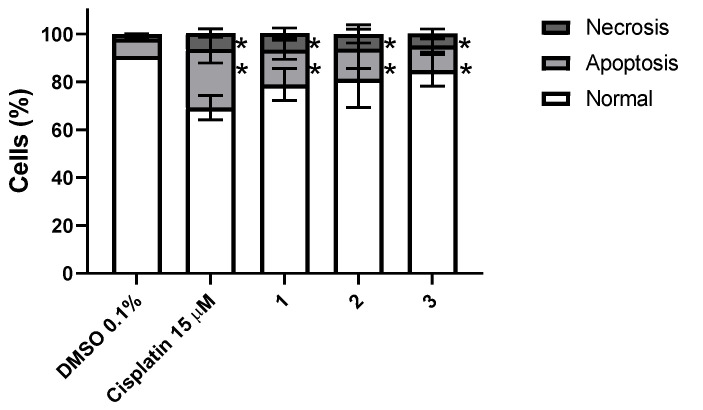
Apoptosis induction in the HCT116 cell line evaluated by flow cytometry after 48 h exposure to IC_50_ concentrations of complexes **1**–**3**. DMSO 0.1% (*v*/*v*) was used as a negative control and Cisplatin 15 μM was used as a positive control. The results presented are mean ± standard deviation of three independent assays. An asterisk indicates a *p*-value inferior to 0.05.

**Figure 13 molecules-26-06364-f013:**
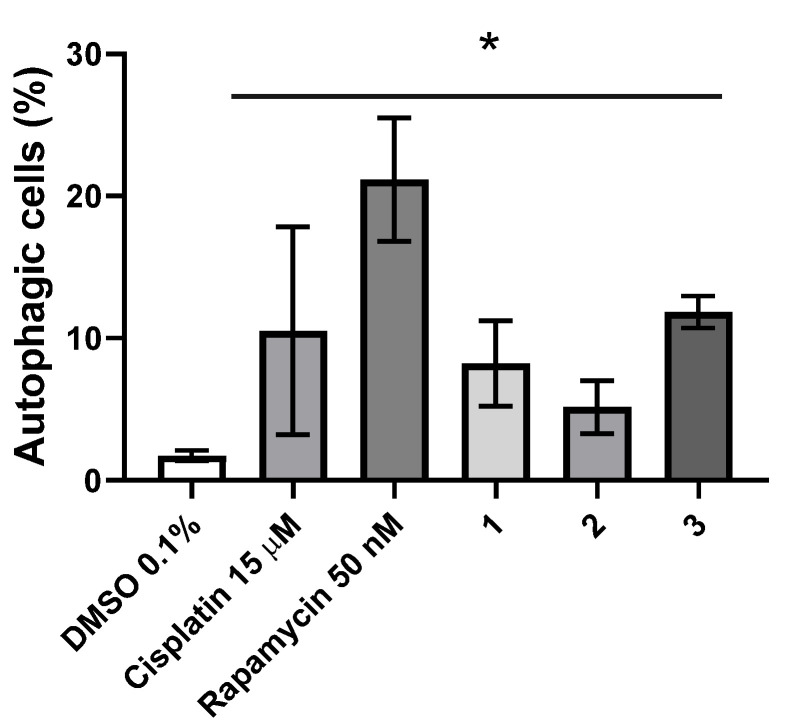
Autophagy induction in the HCT116 cell line evaluated by flow cytometry after 48 h exposure to IC_50_ concentrations of complexes **1**–**3**. DMSO 0.1% (*v*/*v*) was used as a negative control and Cisplatin 15 μM and rapamycin 50 nM were used as positive controls. The results presented are mean ± standard deviation of three independent assays. An asterisk indicates a *p*-value inferior to 0.05.

**Figure 14 molecules-26-06364-f014:**
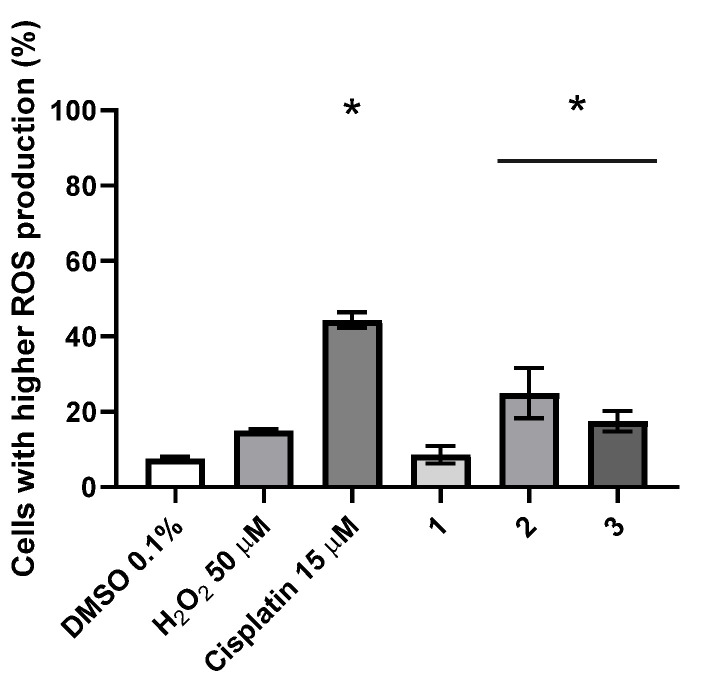
Intracellular ROS production in the HCT116 cell line after 48 h exposure to IC_50_ concentrations of complexes **1**–**3,** evaluated by flow cytometry. DMSO 0.1% (*v*/*v*) was used as a negative control and Cisplatin 15 μM and H_2_O_2_ 50 μM were used as positive controls. The results presented are mean ± standard deviation of three independent assays. An asterisk indicates a *p*-value inferior to 0.05.

**Figure 15 molecules-26-06364-f015:**
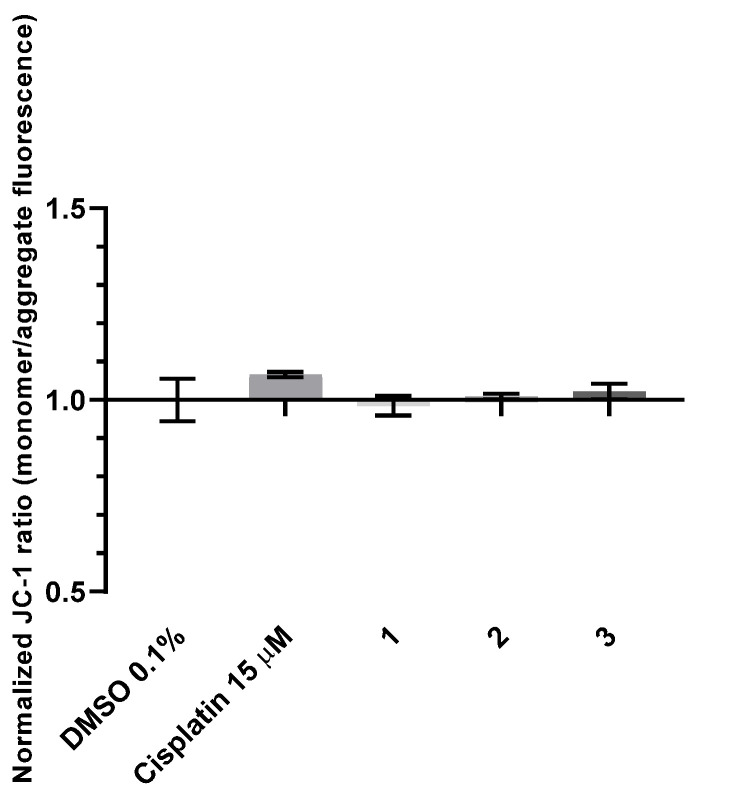
Alterations of the mitochondrial membrane potential in the HCT116 cell line after 48 h exposure to IC_50_ concentrations of complexes **1**–**3,** evaluated by flow cytometry. DMSO 0.1% (*v*/*v*) was used as a negative control and Cisplatin 15 μM was used as a positive control. The results presented are mean ± standard deviation of three independent assays.

**Figure 16 molecules-26-06364-f016:**
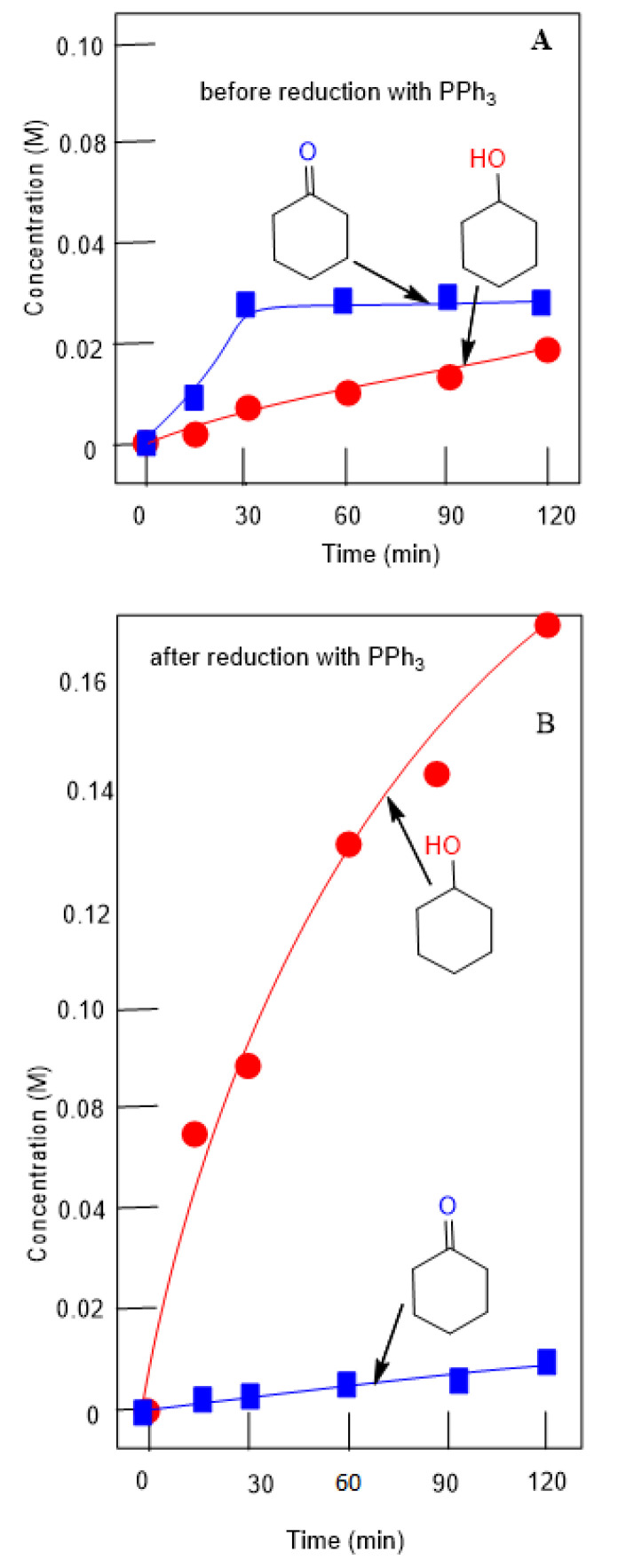
Accumulation of cyclohexanol and cyclohexanone in the oxidation of cyclohexane (0.46 M) with H_2_O_2_ (2.0 M) catalyzed by complex **2** (5 × 10^−4^ M) at 50 °C in acetonitrile. Concentrations of products were measured by GC before (Graph (**A**)) and after (Graph (**B**)) the reduction of the reaction samples with solid PPh_3_.

**Table 1 molecules-26-06364-t001:** Relative IC_50_ values of complexes **1**–**3** in HCT116 and A2780 cancer cell lines and in normal dermal fibroblasts.

Cell Lines	1 (µM)	2 (µM)	3 (µM)
HCT116	12.5 ± 0.63	9.6 ± 0.54	5.9 ± 0.66
A2780	17.5 ± 1.13	20.7 ± 3.11	50.9 ± 2.75
Fibroblasts	15.8 ± 1.49	8.1 ± 2.75	14.5 ± 2.33

**Table 2 molecules-26-06364-t002:** Crystal data and structure refinement for **1**–**2**.

	1	2
Empirical formula	C_22_H_20_N_2_O_3_V	C_22_H_20_N_2_O_3_V
Formula weight	411.34	411.34
*T*, K	295.0(2)	295.0(2)
Wavelength, Å	0.71073	0.71073
Crystal system	Monoclinic	Monoclinic
Space group	*C*2/*c*	*P*2_1_/*c*
Unit cell dimensions, Å and °		
*a*	18.8566(15)	15.6046(11)
*b*	8.2237(4)	8.1475(5)
*c*	13.2105(8)	16.5949(16)
*β*	113.906(5)	117.340(11)
*V*, Å^3^	1872.8(2)	1874.2(3)
*Z*	4	4
*D_c_*, g cm^−3^	1.459	1.458
Absorption coefficient, mm^−1^	0.556	0.555
*F*(000)	852	852
Crystal size, mm	0.283 × 0.137 × 0.055	0.162 × 0.089 × 0.079
*θ* range for data collection °	3.46 to 25.05	3.49 to 25.05
Index ranges	−22 ≤ *h* ≤ 22 −9 ≤ *k* ≤ 9 −15 ≤ *l* ≤ 15	−18 ≤ *h* ≤ 17 −8 ≤ *k* ≤ 9 −19 ≤ *l* ≤ 18
Reflections collected	6154	7065
Independent reflections	1653 [*R*_int_ *=* 0.0221]	3291 [*R*_int_ *=* 0.0479]
Completeness to 2θ	99.7	99.3
Min. and max. transm.	0.712 and 1.000	0.588 and 1.000
Data/restraints/parameters	1653/0/130	3291/0/257
Goodness-of-fit on *F*^2^	1.086	1.002
Final *R* indices [*I* > 2σ(*I*)] *R*1 *wR*2	0.0326 0.0931	0.0489 0.1065
*R* indices (all data) *R*1 *wR*2	0.0355 0.0950	0.0834 0.1202
Largest diff. peak and hole, e Å^−3^	0.50 and −0.24	0.34 and −0.32
CCDC number	1971585	1971586

## Data Availability

Crystallographic data for **1** and **2** have been deposited with the Cambridge Crystallographic Data Center, CCDC 1971585–1971586. Copies of this information may be obtained free of charge from the Director, CCDC, 12 Union Road, Cambridge CB2 1EZ, UK (Fax: +44-1223-336033; e-mail: deposit@ccdc.cam.ac.uk or www.ccdc.cam.ac.uk, 12 December 2019).

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
