# Peer review of "Vanadium(IV) Complexes with Methyl-Substituted 8-Hydroxyquinolines: Catalytic Potential in the Oxidation of Hydrocarbons and Alcohols with Peroxides and Biological Activity"

_molecules, 2021, doi:10.3390/molecules26216364_

Round 1

Reviewer 1 Report

This manuscript describes synthesis, biological and catalytic activity   of three oxovanadium(IV) complexes with the methyl-substituted 8- hydroxyquinolines:   [VO(2,6-(Me)2-quin)2] (1), [VO(2,5-(Me)2-quin)2] (2) and [VO(2-Me-quin)2] (3). The crystal structures of complexes   1 and 2 are determined by single-crystal X-ray diffraction analysis. All experiments in this work are competently performed and described. Overall manuscript is well-written and organized. The presented results could have an importance for chemists dealing in the field of synthesis and biological evaluation of vanadium(IV) complexes with their potential use in medicine as antitumor agents. Therefore, I suggest the acceptance of this manuscript for publication in the journal. I have a few minor remarks which should be considered in the revised manuscript.

1) The oxidation of vanadium should be written in the title of the manuscript. Along with the name of the metal ion, its oxidation state should be always written in brackets.

2) In abstract it should be written what is “quin” in the empirical formula of the investigated complexes.

3) Instead of complexes 1-2 it should be 1 and 2 complexes.  This should be followed throughout the manuscript. Also, instead of complexes 1, 2 and 3 it should be complexes 1-3.

4) In some cases, the complex’ number should be written in bold. 5) What is probability level of thermal ellipsoids in Figure 1. This should be written in the figure caption.

Author Response

List of Changes and Authors’ Replies to Referees

We thank the Reviewers for their comments and suggestions which helped us to improve the manuscript.

Answers to the comments of Reviewer 1

Reviewer 1 wrote: This manuscript describes synthesis, biological and catalytic activity   of three oxovanadium(IV) complexes with the methyl-substituted 8- hydroxyquinolines:   [VO(2,6-(Me)2-quin)2(1), [VO(2,5-(Me)2-quin)2(2) and [VO(2-Me-quin)2(3). The crystal structures of complexes   and are determined by single-crystal X-ray diffraction analysis. All experiments in this work are competently performed and described. Overall manuscript is well-written and organized. The presented results could have an importance for chemists dealing in the field of synthesis and biological evaluation of vanadium(IV) complexes with their potential use in medicine as antitumor agents. Therefore, I suggest the acceptance of this manuscript for publication in the journal. I have a few minor remarks which should be considered in the revised manuscript.

Reviewer’s Remark: The oxidation of vanadium should be written in the title of the manuscript. Along with the name of the metal ion, its oxidation state should be always written in brackets.

Authors’ response: It has been done: “Vanadium(IV) Complexes with Methyl-substituted 8-Hydroxyquinolines: Catalytic Potential in the Oxidation of Hydrocarbons and Alcohols with Peroxides and Biological Activity”.

Reviewer’s Remark: In abstract it should be written what is “quin” in the empirical formula of the investigated complexes.

Authors’ response: It has been done: Methyl-substituted 8-hydroxyquinolines (Hquin) were successfully used to synthetize fivecoordinated oxovanadium(IV) complexes - [VO(2,6-(Me)2-quin)2] (1), [VO(2,5-(Me)2-quin)2] (2) and [VO(2-Me-quin)2] (3).

Reviewer’s Remark:   Instead of complexes 1-2 it should be and 2 complexes.  This should be followed throughout the manuscript. Also, instead of complexes 1, 2 and it should be complexes 1-3.

Authors’ response: It has been done.

Reviewer’s Remark:   In some cases, the complex’ number should be written in bold.

Authors’ response: It has been done.

Reviewer’s Remark: What is probability level of thermal ellipsoids in Figure 1. This should be written in the figure caption.

Authors’ response: Displacement ellipsoids are drawn at the 50% probability level. The information has been added to the figure caption.

Reviewer 2 Report

This work describes the synthesis, catalytic and biological properties of three oxidovanadium(IV) complexes derived from methyl-substituted 8-hydroxyquinolines ligands. Overall, the paper is suitable for publication in Molecules but some points can be addressed before it will be accepted:

  • Which solvent was used for catalytic assay? Is it the same in which the EPR spectra were determined? If not, EPR spectra in the used solvent should be added. In addition, the results concerning blank reactions (to avoid that solvent alone interferes in the catalytic reaction) are missing.
  • How were the concentrations of the catalysts selected? If the solvent for the complexes is CH3CN, are the complexes stable in CH3CN solution?
  • The authors do not indicate for any of the experiments the statistical procedure used: errors, number of experiments carried out for each of the conditions of the experimental determinations.
  • What about the recovering of the catalyst? Does it continue to be active for further catalytic cycles?
  • Were the starting material (VO(acac)2) tested as catalysts?
  • There are more examples of oxidovanadium(IV) complexes used in cycloalkane activation. Authors should be able to draw a brief comparison table with previously published examples to distinguish their novel results.

Author Response

List of Changes and Authors’ Replies to Referees

We thank the Reviewers for their comments and suggestions which helped us to improve the manuscript.

Answers to the comments of Reviewer 2

Reviewer 2 wrote: This work describes the synthesis, catalytic and biological properties of three oxidovanadium(IV) complexes derived from methyl-substituted 8-hydroxyquinolines ligands. Overall, the paper is suitable for publication in Molecules but some points can be addressed before it will be accepted.

Reviewer’s Remark: Which solvent was used for catalytic assay? Is it the same in which the EPR spectra were determined? If not, EPR spectra in the used solvent should be added.

Authors’ response: The catalytic studies were performed in acetonitrile. EPR spectra in acetonitrile solvent has been added to ESI.

Reviewer’s Remark: In addition, the results concerning blank reactions (to avoid that solvent alone interferes in the catalytic reaction) are missing.

Authors’ response: Since some of us have been studying the catalytic activity of vanadium complexes since 1993, all the data on blank experiments were published in earlier works [88, 92–95,110]. Some references have been added [92–95].

Reviewer’s Remark: How were the concentrations of the catalysts selected? If the solvent for the complexes is CH3CN, are the complexes stable in CH3CN solution?

Authors’ response: We used the optimal concentration of the catalyst (5·10-4M), since with a decrease in the concentration to 1×10-4 M, the reaction rate decreases threefold, and an increase in the catalyst concentration to 1×10-3 M increases the reaction rate insignificantly. The catalyst’s concentration has been given in the captions of Figures 6-9. UV-Vis spectra of 1, 2 and 3 in CH3CN  (10-4 M) collected once every two hours over 24 h at room temperature have been added to the ESI.

Reviewer’s Remark: The authors do not indicate for any of the experiments the statistical procedure used: errors, number of experiments carried out for each of the conditions of the experimental determinations.

Authors’ response: Number of experiments carried out for each of the conditions of the experimental determinations were 3 or 4. We calculated the experimental error using the methodology given in the book by [A book Кассандрова О.Н., Лебедев В.В. Обработка результатов наблюденийМ.: Наука1970. — 104 с.].The errors are from 10 to 20% for various experiments.

Reviewer’s Remark: What about the recovering of the catalyst? Does it continue to be active for further catalytic cycles?

Authors’ response: The catalyst changes irreversibly during the reaction, which often occurs in homogeneous catalysis. At the end of the reaction, a precipitate forms. No further reaction was observed after the addition of a new portion of hydrogen peroxide.

Reviewer’s Remark: Were the starting material (VO(acac)2) tested as catalysts?

Authors’ response: We set ourselves the task of investigating the catalytic properties of new vanadium complexes with redox-active ligands in our work; therefore, we did not investigate the catalytic properties of other complexes. However, the catalytic potential of VO(acac)2 in the oxidation of cyclohexane by hydrogen peroxide in acetonitrile was investigated by Alexander Pokusta and co-workers. The results have been added to Table S8, showing the comparison of the catalytic potential of examined complexes 1-3 with the precursor (VO(acac)2) and related vanadium(IVorV) systems. As reported by Pokusta the catalytic performace of VO(acac)2 was very low, but it can be significantly increased by the use of oxalic acid as as promoter [Pokutsa, A.; Bloniarz, P.; Fliunt, O.; Kubaj, Y.; Zaborovskyi, A.; Pacześniak, T. Sustainable oxidation of cyclohexane catalyzed by a VO(acac)2-oxalic acid tandem: the electrochemical motive of the process efficiency. RSC Adv. 2020, 10, 10959–10971; Pokutsa, A.;  Kubaj, Y.; Zaborovskyi, A.; Sobkowiak, A., Muzart, J. Oxalic acid-improved mild cyclohexane oxidation catalyzed by VO(acac)2: non-radical versus radical mechanism. Reac. Kinet. Mech. Cat. 2017, 122, 757–774.]

Round 2

Reviewer 2 Report

The required corrections have been addressed by the authors, and now the work is of sufficient quality to be published in Molecules